# Rapid and Non-Invasive Assessment of Texture Profile Analysis of Common Carp (*Cyprinus carpio* L.) Using Hyperspectral Imaging and Machine Learning

**DOI:** 10.3390/foods12173154

**Published:** 2023-08-22

**Authors:** Yi-Ming Cao, Yan Zhang, Shuang-Ting Yu, Kai-Kuo Wang, Ying-Jie Chen, Zi-Ming Xu, Zi-Yao Ma, Hong-Lu Chen, Qi Wang, Ran Zhao, Xiao-Qing Sun, Jiong-Tang Li

**Affiliations:** 1Key Laboratory of Aquatic Genomics, Ministry of Agriculture and Rural Affairs, Beijing Key Laboratory of Fishery Biotechnology, Chinese Academy of Fishery Sciences, Beijing 100041, China; caoyiming@cafs.ac.cn (Y.-M.C.); zhangy@cafs.ac.cn (Y.Z.); styuwork@163.com (S.-T.Y.); chenhongluz@cafs.ac.cn (H.-L.C.); wangqi@cafs.ac.cn (Q.W.); zhaoran@cafs.ac.cn (R.Z.); sunxiaoqing@cafs.ac.cn (X.-Q.S.); 2Chinese Academy of Agricultural Sciences, Beijing 100181, China; 3National Demonstration Center for Experimental Fisheries Science Education, Shanghai Ocean University, Shanghai 201306, China; 18631836881@163.com (K.-K.W.); cyjttkl@163.com (Y.-J.C.); xuziming0916@163.com (Z.-M.X.); zql3703700@163.com (Z.-Y.M.)

**Keywords:** common carp, hyperspectral imaging, texture, machine learning, visualization

## Abstract

Hyperspectral imaging (HSI) has been applied to assess the texture profile analysis (TPA) of processed meat. However, whether the texture profiles of live fish muscle could be assessed using HSI has not been determined. In this study, we evaluated the texture profile of four muscle regions of live common carp by scanning the corresponding skin regions using HSI. We collected skin hyperspectral information from four regions of 387 scaled and live common carp. Eight texture indicators of the muscle corresponding to each skin region were measured. With the skin HSI of live common carp, six machine learning (ML) models were used to predict the muscle texture indicators. Backpropagation artificial neural network (BP-ANN), partial least-square regression (PLSR), and least-square support vector machine (LS-SVM) were identified as the optimal models for predicting the texture parameters of the dorsal (coefficients of determination for prediction (*r*_p_) ranged from 0.9191 to 0.9847, and the root-mean-square error for prediction ranged from 0.1070 to 0.3165), pectoral (*r*_p_ ranged from 0.9033 to 0.9574, and RMSEP ranged from 0.2285 to 0.3930), abdominal (*r*_p_ ranged from 0.9070 to 0.9776, and RMSEP ranged from 0.1649 to 0.3601), and gluteal (*r*_p_ ranged from 0.8726 to 0.9768, and RMSEP ranged from 0.1804 to 0.3938) regions. The optimal ML models and skin HSI data were employed to generate visual prediction maps of TPA values in common carp muscles. These results demonstrated that skin HSI and the optimal models can be used to rapidly and accurately determine the texture qualities of different muscle regions in common carp.

## 1. Introduction

The textural traits of fish, including gumminess, springiness, cohesiveness, resilience, hardness, brittleness, adhesiveness, and chewiness, are the most important traits in the aquaculture industry, and they affect the production process and the commercial value of fish [1,2,3]. Developing the fillet textual assessment method is beneficial for measuring the textual traits of fish-processed products [4]. Traditional fish textual assessment methods include measurements using a texture analyzer [5,6]. However, these methods are laborious and might destroy the integrity of the products. Therefore, there is an immediate requirement to construct an efficient and non-destructive method to detect the muscle texture of processed fish.

Recently, hyperspectral imaging (HSI) has become an alternative analytical approach that provides the benefits of rapid and non-destructive detection [4,7,8,9,10,11]. HSI combines image and spectral techniques to obtain both “spatial” and “spectral” information containing the sample [9,10,11,12]. Another feature of HSI is the ability to generate visual distribution maps of measured indicators to allow for the prediction and quantification of the composition of the sample and to determine their position on the sample surface [6,13]. Moreover, artificial intelligence and machine learning (ML) models can be used for prediction and modeling in the food industry [14]. With spectral images, HSI has been widely applied to evaluate the traits of meat products, including color, surface defects, damage, texture, water-holding capacity, flavor, freshness, and ripeness [4,15,16,17,18,19,20,21]. Ma et al. used HSI based on 400–1000 nm wavelengths to predict the different textural parameters of grass carp fillets during vacuum freeze-drying [4]. They predicted the Warner–Bratzler shear force, hardness, gumminess, and chewiness of fillets with prediction coefficients ranging from 0.79 to 0.87. ElMasry et al. predicted beef tenderness using hyperspectral imaging with a model based on partial least squares (PLS), showing a detection coefficient of 0.83 and a cross-validation narrative of 0.75 [17]. Zhou et al. predicted six texture parameters of silver carp muscle using HSI and ML methods, with coefficients ranging from 0.83 to 0.95 [8]. In addition, He et al. found that the SPA-LS-SVM prediction model and HSI had a prediction coefficient of 0.905 for the tenderness of salmon fillets [22]. These studies demonstrate that HSI and ML methods provide reliable solutions to measure processed fish textures.

In fish breeding, high-quality textures can provide fillets that are suitable for downstream processing and satisfy the consumer’s taste. The traditional textual method requires the cut of fish muscle and is lethiferous [5,6]. Compared with the traditional textual method, HSI and ML methods have a non-destructive advantage, as they allow for the detection of the texture of live fish muscle. In the current literature, the majority of researchers have investigated the quality of fillets rather than intact fish using HSI, meaning that the spectra were usually obtained from the meat mass [16,23,24]. However, the application of HSI and ML methods to measure the live fish muscle has been less studied.

Common carp (*Cyprinus carpio*), an allotetraploid fish [25], is one of the most important freshwater-farmed fish in the world. Therefore, the aim of this study was to develop a non-invasive method in which skin HSI and ML are combined to detect the textual parameters of live fish muscle. We first acquired the skin HSI data of 387 scaled and live common carp with a hyperspectral imaging system at 400–1000 nm. Then, we measured the texture profiles of four corresponding muscle regions of each fish. The specific objectives of this study were to (1) utilize preprocessing methods to achieve spectral preprocessing and characteristic wavelength selection; (2) determine the optimal wavelengths that are most useful for the prediction of texture profile analysis (TPA) within the muscle of common carp; (3) determine the optimal relationship between the skin HSI data and muscle texture parameters using six ML methods and incorporate the skin hyperspectral index; and (4) apply the optimal model for the visualization of the distribution of muscle texture parameters.

## 2. Materials and Methods

### 2.1. Ethics Statement and Sampling

We performed this study following the recommendations of the Animal Care and Use Committee of the Chinese Academy of Fishery Sciences. In 2022, 387 live common carp (one-year-old, ~607.09 g average weight) were collected from the Chinese Academy of Fishery Sciences experimental fish farm (Fangshan, Beijing, China). These samples were previously cultivated at one pond and fed the same commercial diet (Tongwei, China). The sources of protein and lipids in commercial diet are fish meal, soybean meal, cottonseed meal, rapeseed meal, and soybean oil. The proximate compositions of the commercial diet (% dry matter) consist of 30% protein, 5% lipid, 12.5% moisture, and 15% ash. For each common carp, we selected four regions, including the dorsal, pectoral, abdominal, and gluteal regions.

### 2.2. Acquiring the Skin Hyperspectral Images of Four Regions of Live Common Carp

For each region of one live common carp, the scales on one side were removed. We scanned the skin of four regions of one scaled common carp with a hyperspectral imager (FigSpec Hyperspectral Camera FS-13, FigSpec Technology (Zhejiang) Co., Ltd., Hang Zhou, China). The detected wavelengths ranged from 400 to 1000 nm with the following parameters: a resolution of 2.5 nm, an exposure period of 150 ms, and the lens type of C-Mount. The imaging speed was 128 Hz in the full-wavelength range, and the scanning speed was 30 row/s. Before acquiring the hyperspectral images of each sample, the distance and intensity of the illumination source were adjusted to ensure the clearness of the acquired images.

### 2.3. Processing the Skin HSI

HSI enables the collection of signals from samples, as well as the environment, instruments, and other non-sample factors. To eliminate the signals from the non-sample factors, all HSI data were input into the reflectance calibration procedure. Briefly, the raw data were calibrated with black and white correction. The white balanced image (***W***) was obtained by collecting the reflectance value from the Teflon white surface, while the dark image (***D***) was acquired by turning off the illumination source and collecting the hyperspectral data when the lens was completely covered with its cap. The calibration image (***I***) was calculated using the following equation:$$I=\frac{I_{0}-D}{W-D}\times 100$$
where I represents the corrected reflectance hyperspectral image in a unit of relative reflectance (%); $I_{0}$ represents the raw hyperspectral image; D stands for the dark image (0% reflectance); and W is the white reference image (100% reflectance) [26].

Then, we used the Savitzky–Golay (SG) smoothing method to preprocess the images and eliminate the putative effects from the sampling environment and instruments [27]. We selected a region of interest (ROI) to represent each skin region with the ROI function of Environment in the Visualizing Images software (ENVI v5.3, Exelis Visual Information Solutions, Inc., Boulder, CO, USA) [5]. The size of an ROI was 200 pixels × 200 pixels. For each wavelength, the average spectrum of an ROI was calculated by averaging the spectra of all pixels. The reflectance values of all pixels were averaged at each wavelength variable to obtain an average value representative of each sample.

### 2.4. Selecting the Optimal Wavelength

One HSI dataset contained the spectral information of samples from 400 nm to 1000 nm, simultaneously. However, certain wavelengths had redundant data, resulting in the time-consuming processing of HSI data [28]. Therefore, it is necessary to eliminate wavelengths containing redundant and irrelevant information to optimize the texture profiles for data analysis samples using the wavelength/variant selection of hyperspectral data [23]. Regarding wavelength selection for HSI analysis, the regression coefficient (RC) is commonly utilized [29,30]. We utilized the RC to determine the optimal wavelength that contributed the most to the prediction of TPA values in common carp muscle. In the calculation of RC, the optimal wavelength is chosen by computing the β-coefficient from the full-wavelength PLSR model. The wavelength with the highest absolute value of the β-coefficient is considered to be the optimal wavelength [30]. The program for RC was operated in MATLAB 2021a software (The MathWorks Inc., Natick, MA, USA).

### 2.5. Measuring the Texture Indicators of Common Carp Muscle

We extracted the muscle corresponding to each of the four skin regions. The muscle size was 20 mm × 20 mm × 15 mm. Eight texture indicators of the muscle, including gumminess, springiness, cohesiveness, resilience, hardness, brittleness, adhesiveness, and chewiness, were measured using a texture analyzer (TA.XTC-18, Baosheng, Shanghai, China) and a TA/36 cylindrical probe. The measurement speed was 2 mm/s, and the trigger force was 5 N. The compressive deformation of one sample was set to 60%. These eight texture indicators were derived from the TPA curves of each sample, and the TPA parameters listed above were calculated using Bourne’s technique [31].

For each texture indicator, to examine whether there were significant differences among four muscle regions, we measured the distances with PCA analysis using Tassel 5.0 [32]. The Spearman correlation coefficient of the contents of any two indicators in four samples was calculated using the R ‘cor. test’ function in the R software (version 4.0.2).

### 2.6. Estimating the Muscle Texture Indicators with the Skin HSI Data

With the processed skin HSI data in each region, we used six machine learning (ML) methods to estimate the texture indicators of the muscle in the corresponding region. The methods included partial least-square regression (PLSR), the interval partial least-square method (iPLS), the synergy interval partial least-square method (SiPLS), backward interval partial least squares (BiPLS), least-square support vector machines (LS-SVM), and backpropagation artificial neural network (BP-ANN).

PLSR projects the predictor variables and observable variables into a new feature space to build a linear regression model [33]. PLSR decomposes the independent variable X and the dependent variable Y into several X-scores (T) and constructs the PLSR model. Herein, the observed variables were the cross-validation performed to minimize the error between the predicted and the observed response values.

In the iPLS algorithm, the full spectral region is divided into smaller equidistant subintervals, and a PLS regression model is generated based on each subinterval. The best intervals and principal component scores are selected based on the principle of the lowest root-mean-square error for the calibration (RMSEC) value [34].

The SiPLS algorithm is a modified iPLS where the full spectral region is divided equally into subintervals. The combination with the lowest RMSEC value is selected [34].

The BiPLS algorithm divides the whole spectral region into N subintervals of equal width and performs PLS regression, each interval is omitted in turn, and the worst RMSEC value is obtained in the modeling; the subintervals continue to be removed until the lowest RMSEC value is obtained [34].

LS-SVM uses the radial basis kernel function (RBF), a non-linear function that reduces the complexity of the training process [35]. The regularization parameter gamma (γ) and the kernel parameter (σ^2^), which can reduce the complexity, represent the width of the RBF kernel. To achieve high prediction accuracy, we performed the simulations of these two parameters, the values of which ranged from 0 to 1000 [6].

In BP-ANN models, an error-reversal propagation algorithm is used to train multilayer feedforward neural networks [36]. A BP-ANN, with an input layer, a hidden layer, and an output layer was established. Moreover, the transfer function, learning function, and training function were employed. The maximum training step was set to 1000, the learning goal was e^−5^, and the learning rate and momentum factor were 0.01.

### 2.7. Evaluating the Accuracies of Six ML Models

The predictive accuracy of each ML model was assessed with multiple parameters, including coefficients of determination for calibration (*r*_c_) and prediction (*r*_p_), RMSEC, and the root-mean-square error for calibration and prediction (RMSEP) [37]. The *r*_c_ and *r*_p_ values were calculated as follows:$$r_{C}=\sqrt{\frac{\sum_{i=1}^{n_{c}}{\left({{\hat{y}}_{i}-y}_{i}\right)}^{2}}{\sum_{i=1}^{n_{c}}{\left({\hat{y}}_{i}-y_{c}\right)}^{2}}}$$
$$r_{P}=\sqrt{\frac{\sum_{i=1}^{n_{p}}{\left({\hat{y}}_{i}{-y}_{i}\right)}^{2}}{\sum_{i=1}^{n_{p}}{\left({\hat{y}}_{i}{-y}_{p}\right)}^{2}}}$$
where ${\hat{y}}_{i}$ and $y_{i}$ represent the predicted and measured TPA values, respectively; $n_{c}$ and $n_{p}$ represent the number of samples in the calibration and prediction sets, respectively.

The RMSEC and RMSEP were calculated as follows:$$RMSEC=\sqrt{\frac{1}{N-1-R}\times \sum_{i=1}^{N}{\left(y_{i}^{ref}-y_{i}\right)}^{2}}$$
where N is the number of samples, R is the number of factors of the model, $y_{i}^{ref}$ is the reference value of the sample, and i and $y_{i}$ are the predicted values of the sample.
$$RMSEP=\sqrt{\frac{1}{N}\times \sum_{i=1}^{N}{\left(y_{i}^{ref}-y_{i}\right)}^{2}}$$

Herein, for each texture indicator, $y_{i}^{ref}$ was the observed value in the common carp muscle, while $y_{i}$ was the predicted value with one ML method and the reflectance values of corresponding skin HSI. The lower RMSEC and RMSEP values indicated a smaller difference between the predicted texture indicator and the observed indicator. A good ML model was expected to have high *r*_c_ and *r*_p_ but low RMSEC and RMSEP values [38].

### 2.8. Visualizing the Images of TPA Values

TPA values were distributed varyingly in different muscle regions of the fillets [39], which resulted from the irregular distribution of lipids and protein in different muscle regions. To examine the differences among the TPA values in different muscle regions of one fish, distribution maps of TPA values were constructed to improve insight into the muscle texture of common carp. The optimal calibration model constructed by applying the spectra of the optimal wavelengths following RC selection was employed to generate new distribution maps of TPA values. Linear color scales are presented in the figure by visualizing the distribution maps, and the different colors in the color scales represent the predicted TPA parameter values in the fillets, thus facilitating the identification and capture of the variations in muscle TPA values by observing different color distributions. All the calculation and visualization procedures were implemented in programs operating in ENVI 5.3 (Exelis Visual Information Solutions, Inc., Boulder, CO, USA) and MATLAB 2021a software (The MathWorks Inc., Natick, MA, USA) [5].

## 3. Results

### 3.1. Spectral Features of the Skin of Scaled Common Carp

The spectral features of the four skin regions of common carp were distinct (Figure 1). For all regions, the spectrum at 430 nm had the lowest reflectance values. The distributions of the reflectance values in the gluteal, pectoral, and ventral skins were different from that of the dorsal skin. In general, for the former three skin regions, the reflectance values gradually increased at 430–600 nm. The values reached the plateau phase at 600–780 nm and fell at 780–970 nm. Finally, the values increased after 970 nm. However, the reflectance values in the dorsal skin gradually increased from 430 nm to 1000 nm. The reflectance value of the gluteal skin at each wavelength was higher than those of the other skins. The reflectance values of the pectoral and ventral skin ranked second and third. The values of the dorsal skin were the lowest. The distinct distributions and the levels of the reflectance values among the four skins might indicate the different features of the four skin regions or the affiliated tissues.

### 3.2. Texture Diversities of Common Carp Muscles

We obtained eight textural parameters of four muscle regions of 387 common carp (Appendix A). In the dorsal region, the first two principal components (PCs) explained 73.62% and 12.73% of all variances, respectively (Appendix A). Intriguingly, the examined samples were grouped into two different clusters, suggesting different textural profiles among samples. Similar phenomena of two clusters were observed in the PCA analysis using the indicators of the pectoral, abdominal, and gluteal regions, respectively (Appendix A). The hardness of the gluteal region (median = 2010.53) was significantly higher than that of the other three regions (Figure 2 and Appendix A). The pectoral hardness was the lowest. Intriguingly, except for the cohesiveness (Appendix A) and adhesiveness indicators (Appendix A), the other five indicators in the gluteal region were also significantly higher than those of the other three regions (Appendix A). However, the adhesiveness parameter in this region was the lowest. These data revealed the different texture features of the four muscle regions.

The numbers of texture indicator pairs with a significant correlation were 17, 14, 9, and 7 in the abdominal, pectoral, dorsal, and gluteal regions, respectively (Appendix A). Only resilience had a significantly positive correlation with the springiness (coefficients ranging between 0.728 and 0.990) and cohesiveness (coefficients ranging between 0.148 and 0.473) in all four regions, respectively. The cohesiveness was also significantly positively correlated with chewiness (coefficients ranging between 0.12 and 0.951) in all four regions. Six texture indicator pairs had significant correlations among the three regions, including four positively correlated pairs and two negatively correlated pairs. These data suggested that the most significant correlation was not consistent in the four regions.

### 3.3. Accurate Prediction of Muscle Texture Profiles Based on the Full Spectral Range

Since the texture profiles of the four muscle regions were different, we tried to determine whether it is possible to predict the muscle texture profiles. We used all reflectance values of the skin HSI data to predict the corresponding muscle texture indicators with different ML methods. For one texture indicator of one region, we only retained the prediction with the highest *r*_p_ for the downstream analysis (Appendix A). Predicting the chewiness of the four muscle regions had the highest *r*_p_ (from 0.9555 to 0.9836). The overall prediction accuracies of the gumminess (*r*_p_ from 0.9234 to 0.9863) and cohesiveness (*r*_p_ from 0.8952 to 0.9224) of the four muscle regions ranked second and third, respectively. The overall prediction accuracies of the hardness and adhesiveness indicators were also higher than 0.88 for all four regions. Among all the best predictions, the prediction accuracy of dorsal springiness was the lowest, with only 0.5612.

The BP-ANN, LS-SVM, and PLSR models were the best three methods to predict the muscle TPA parameters (Table 1, Table 2, Table 3 and Table 4). The BP-ANN method had the best calibration results mainly for gumminess, chewiness, cohesiveness, hardness, and adhesiveness, including dorsal gumminess (0.9863), pectoral gumminess (0.9620), dorsal chewiness (0.9673), pectoral chewiness (0.9555), abdominal chewiness (0.9690), gluteal chewiness (0.9836), dorsal cohesiveness (0.9224), pectoral cohesiveness (0.9306), abdominal hardness (0.9401), and gluteal adhesiveness (0.9303). LS-SVM had the highest prediction accuracy for adhesiveness, including dorsal adhesiveness (0.9206) and abdominal adhesiveness (0.9206). PLSR had the optimal prediction effect mainly for gumminess, cohesiveness, and hardness, including abdominal chewiness (0.9318), gluteal chewiness (0.9836), abdominal gumminess (0.9318), gluteal gumminess (0.9234), and pectoral hardness (0.9033).

### 3.4. Accurate Prediction of the Muscle Texture Profiles Based on the Optimum Wavelengths

Equivalent calibration results were obtained based on the optimal wavelengths compared with full wavelengths. This is because the optimal wavelengths carry the most important information relevant to the determination. Some peaks and valleys (positive and negative relationships with the TPA parameters) were selected at certain wavelengths, and the selection of optimal wavelengths was successfully conducted for the eight texture parameters using the RC method. We selected 60 to 114 optimal wavelengths of the dorsal, pectoral, abdominal, and gluteal skin regions to predict the muscle textures, respectively (Figure 3).

For one texture indicator of one region, we only retained the prediction with the highest *r*_p_ for the downstream analysis (Appendix A). In general, the prediction accuracies based on the optimal wavelengths were equal to those based on the full-wavelength range (Table 5, Table 6, Table 7 and Table 8). Using the values in the range of the full wavelength, nine predictions had accuracies lower than 0.85. The accuracies of these regions were still lower than 0.85 using the optimal wavelengths. The remaining regions had accuracies over 0.85 using either the values of the full wavelength or the ones of the optimal wavelength. Moreover, the absolute prediction differences between the full wavelength values and the optimal wavelength values ranged from 5.93% to 15.20%, showing that the wavelength selection could make the reduced models more stable and robust.

Compared with the prediction of TPA values based on the full-wavelength range, the *r*_p_ values of four TPA parameters (cohesiveness, hardness, springiness, and resilience) in the dorsal region were enhanced using the optimal wavelength, with the increase ranging from 0.0009 to 0.1014. The remaining four values slightly decreased. In the pectoral muscle, the *r*_p_ values of five TPA parameters, including adhesiveness, were increased by 0.0475. In the abdominal muscle region, three indicators (gumminess, springiness, and resilience) had improved *r*_p_ values. However, in the gluteal muscle region, only the accuracies of the gumminess raised by 0.0105. These results indicate that for, the former three regions, the prediction using the optimal wavelengths would be better than using the full-range wavelengths.

The BP-ANN, LS-SVM, and PLSR models were also the best three methods to predict the muscle TPA parameters. The BP-ANN method had the best calibration tool mainly for chewiness, gumminess, hardness, and adhesiveness, including dorsal chewiness (0.9469), pectoral chewiness (0.9552), abdominal chewiness (0.9776), gluteal chewiness (0.9768), dorsal gumminess (0.9847), pectoral gumminess (0.9574), pectoral hardness (0.9033), dorsal adhesiveness (0.9191), and gluteal adhesiveness (0.9304). LS-SVM had the highest prediction accuracy for gumminess and adhesiveness, including gluteal gumminess (0.9339) and pectoral adhesiveness (0.9370). PLSR had the optimal prediction effect mainly for gumminess, cohesiveness, and hardness, including abdominal gumminess (0.9517), dorsal cohesiveness (0.9367), pectoral cohesiveness (0.9056), abdominal cohesiveness (0.9070), dorsal hardness (0.9298), and abdominal hardness (0.9392).

### 3.5. Visualizing the Texture Parameters

The muscle texture parameters could be accurately predicted with the skin HSI spectra and the corresponding models. Therefore, the skin HSI spectra based on the optimal wavelengths and the above models were used to predict the predicted muscle parameters, which were further converted to the corresponding pixels in the tested samples, and the prediction maps were then generated. Figure 4 displays the visual prediction images of eight texture parameters in the dorsal region. The color variations presented in the test samples are automatically condensed in a linear color bar. The colors correspond to the different texture levels of the samples. The low values are highlighted in blue, and the high values are shown in orange.

In one distribution map, the spots with the same color were discretely distributed. Spots with high values were rare. Even in the same region, the prediction map of each parameter was in general different from the others. These maps reflected the minute texture difference in one region.

## 4. Discussion

Previous works that focused on muscle texture prediction mainly utilized muscle HSI [4,5,8]. One highlight of our work was using the corresponding skin HSI of different muscle regions to estimate the texture features. Although we used the reflectance values of the skin, we found that the skin wavelength distributions were in agreement with the muscle wavelength distributions in a previous study [6]. Moreover, the high prediction accuracies of muscle texture profiles with skin HSI (prediction coefficients >0.9 for the majority of texture parameters) demonstrated that this strategy can be used in practice to detect muscle texture qualities. The prediction results were even higher than those already reported, so this method is feasible [5,8]. Another highlight of our work was performing multiple ML methods to predict muscle texture profiles, which is different from the methods used in previous studies [4]. Our results showed that BP-ANN, LS-SVM, and PLSR were the best three methods to predict the muscle TPA parameters. Each of these three methods was suitable for specific texture indicators in different muscle regions.

The distinct distributions and the levels of the reflectance values among the four skin regions, together with the different texture features of the four muscle regions, indicated the different features of these regions and the corresponding affiliated tissues. The differences were probably due to the differences in the primary chemical composition of the epidermis of the different muscle regions of common carp [40]. When the electromagnetic radiation emitted by light interacts with the internal structures of the sample, the various components of the sample exhibit distinct absorption properties at multiple particular wavelengths [17]. In the absorption, information in the 400–1000 nm spectral regions, overtones, and combinations of fundamental vibrations of functional bonds such as C-H, N-H, O-H, and S-H occur [24]. An interesting spectral trough was detected at around 430 nm, and comparable patterns were observed in the evaluation of total volatile basic nitrogen (TVB-N) and TPA in grass carp, although there have been few studies of this specific wavelength in common carp [4,6]. There was a noticeable and large absorption peak at around 500 nm, which might be associated with the residues of organic dietary items such as soybean meal [41]. Another local absorption at around 780 nm was mostly attributable to a third overtone O-H stretching [42]. The presence of water in fish caused absorption peaks at 980 nm (O-H stretching second overtone) [43].

The prediction coefficients for gumminess, chewiness, cohesiveness, and adhesiveness for different muscle regions ranged from 0.9206 to 0.9863. Wu et al. reported prediction coefficients for textural metrics of salmon fillets using full-wavelength spectroscopy ranging from 0.555 to 0.665 [5]. Chen et al. demonstrated a prediction coefficient of 0.80 for chewiness and RMSEP of 0.942 in beef [44]. Overall, the predictive ability was better than that of previous studies. A possible reason for this is the fact that different structures exhibit different characteristics of light scattering, projection, and reflection [45]. Tissues with denser muscle fibers and softer connective tissue result in better prediction [46]. These studies indicate the complexity of the elements that impact the prediction of meat quality characteristics.

Although the texture parameters were satisfactorily evaluated using full wavelengths, the volume of data and the amount of computation are enormous. Choosing the optimal wavelength can reduce the data dimensions and increase the computational speed of the model. Ma et al. applied the optimal wavelength to build a Warner–Bratzler shear force prediction model in which there was only a slight reduction from 0.8955 to 0.8913, while the number of variables both reduced significantly from 381 to 10 [4]. Our results indicate that the prediction accuracies using the optimal wavelength were not significantly different from those using the full wavelength, and similar results have been observed in previous studies [22,40]. Moreover, 81% of the wavelengths were excluded from the full-spectrum scope (114 compared to 600), indicating that RC was a valuable method of wavelength selection for identifying TPA values in the muscle of common carp. Decreasing the wavelength numbers while ensuring accuracy also reduced the runtime. Moreover, the prediction of TPA parameters such as gumminess, cohesiveness, and chewiness using the optimal wavelength obtained similar results, with *r*_p_ values ranging from 0.91 to 0.98. Using hyperspectral information from the skin and combining it with ML algorithms to predict the TPA of muscles had high accuracy. The prediction results were even higher than those already reported, so this method is feasible [5,8].

Although we obtained more optimized prediction results, the accuracy of the predictions depends on the quality and fitness of the calibration model. The prediction coefficients for springiness and resilience were lower in the present study (ranging from 0.5609 to 0.7613), and similar results were observed in a study of TPA of salmon fillets based on visible and near-infrared spectroscopy [5]. The reason for this unsatisfactory prediction may be due to the fact that springiness and resilience are subject to differences in muscle structure and connective tissue as well as differences in the water and myofibrillar protein content in muscle [47]. Therefore, there are still areas for improvement in HSI techniques, but the accuracy and efficiency of the method will continue to improve with advances in machine learning and spectroscopy.

The implementation of the visualization process is the ultimate but essential step in the HSI technique for texture prediction and will contribute to understanding the changes in TPA values in carp muscles that cannot be detected by the naked eye [4,5,13]. In one muscle region, different colored spots could be easily identified from the TPA distribution map, indicating that common carp muscles have a mixed composition and heterogeneous texture distribution [6,13]. The main reason for the color differences might be due to the distinct distribution of collagen and fat in the muscle [39,44]. The spatial distribution of textural features in common carp muscles can be conveniently observed by reference value distribution maps generated from the HSI of the samples [4]. Traditional methods can only detect a few specific points of the sample and are destructive and time-consuming. Hyperspectral imaging, on the other hand, with its superior spatial information, provides more detailed information for rapid, non-invasive measurement of TPA in common carp muscle.

The traditional method for measuring the muscle texture indicators requires a texture analyzer. The method is destructive and requires much time for muscle preparation and measurement. Our method, integrating skin hyperspectral imaging, the optimal ML method, and a visual prediction map, might provide a promising alternative tool to measure the muscle texture quality. First, this method does not require muscle preparation and is thus rapid and non-destructive. Second, the distribution maps of muscle TPA values are very useful for the meat industry to assess the sensory quality of common carp muscle by simply observing the color of the distribution map. Third, aside from the meat industry, the rapid, non-destructive, and visible features of our method are also helpful for screening common carp for food requirements.

## 5. Conclusions

The possibility of using HSI techniques (400–1000 nm) as a tool for determining the muscle texture profile in scaled common carp was evaluated. The optimal wavelength selected based on the RC data downscaling method with ML methods (BP-ANN, PLSR, and LS-SVM) performed most efficiently in predicting the TPA of different muscle regions in common carp. The results showed excellent performance in predicting gumminess, cohesiveness, adhesiveness, and chewiness. The *r*_p_ ranged from 0.8726 to 0.9847. Moreover, the visualization map of the distribution of TPA values was generated based on the optimal models, which provided further insight into the texture parameters in the common carp muscles. This study illustrated the tremendous potential of hyperspectral imaging technology as a robust and effective tool for the rapid and non-destructive measurement of TPA in different scaled common carp muscle regions. Despite the superior results of this study in predicting muscle texture parameters in common carp, it is still necessary to validate the developed models by applying numerous samples to ensure their reliability. In future studies, using hyperspectral imaging to acquire hyperspectral image data of other species of fish could be attempted for the rapid and non-destructive detection of meat quality.

## Figures and Tables

**Figure 1 foods-12-03154-f001:**
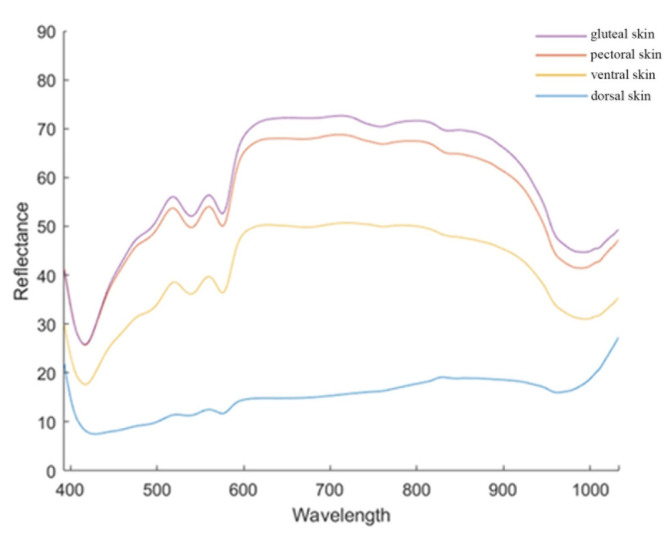
The spectral features of four skin regions of the common carp. The X-axis represents the different wavelengths, the Y-axis represents the reflectance values, and the curves of different colors represent the reflectance of four skin regions.

**Figure 2 foods-12-03154-f002:**
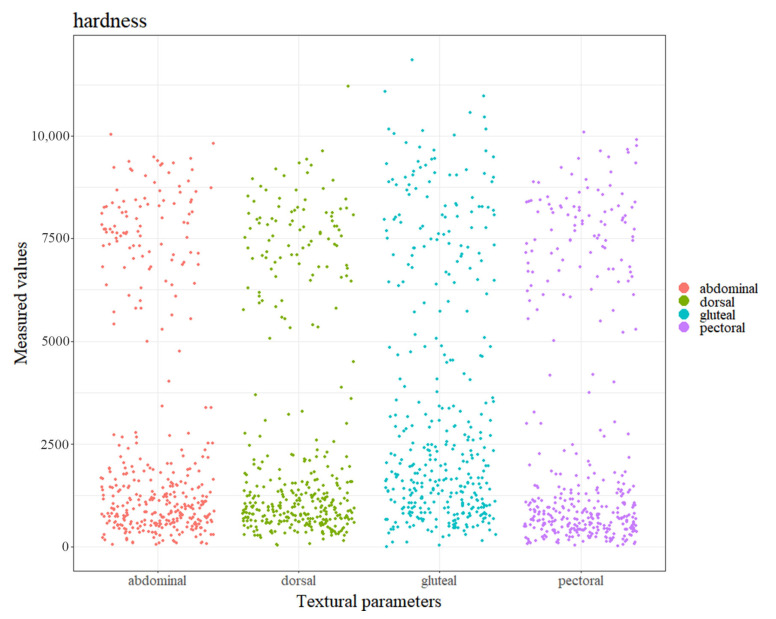
The hardness distributions in the four muscle regions of common carp.

**Figure 3 foods-12-03154-f003:**
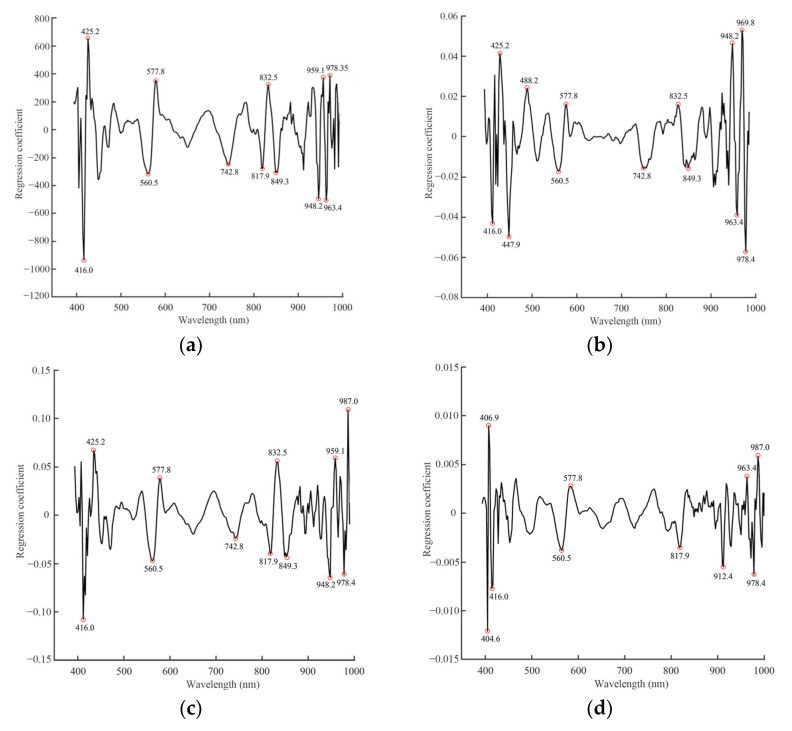
Selection of optimal wavelengths in the dorsal muscle. Regression coefficients method for (**a**) gumminess, (**b**) springiness, (**c**) cohesiveness, (**d**) resilience, (**e**) hardness, (**f**) brittleness, (**g**) adhesiveness, and (**h**) chewiness.

**Figure 4 foods-12-03154-f004:**
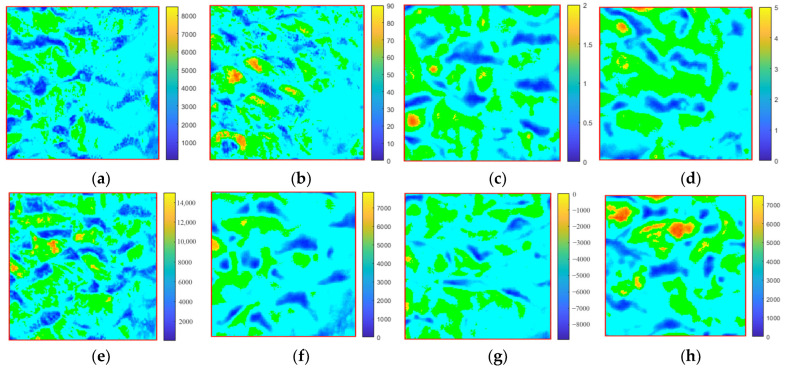
Visualizing the texture parameters of the dorsal muscle of scaled common carp: (**a**) gumminess, (**b**) springiness, (**c**) cohesiveness, (**d**) resilience, (**e**) hardness, (**f**) brittleness, (**g**) adhesiveness, and (**h**) chewiness. The numbers on the Y-axis represent the value of the texture measured, and the color from blue to orange represents the values from low to high.

**Table 1 foods-12-03154-t001:** Predicting the dorsal muscle texture profiles using the skin HSI with the reflectance values in the full-wavelength range.

Texture	Best ML Method	*r* _c_	RMSEC	*r* _p_	RMSEP
Gumminess	BP-ANN	0.9826	0.1439	0.9863	0.0685
Chewiness	BP-ANN	0.9852	0.1325	0.9673	0.1065
Cohesiveness	BP-ANN	0.9694	0.0842	0.9224	0.0735
Adhesiveness	LS-SVM	0.9556	0.1873	0.9206	0.1771
Brittleness	PLSR	0.9025	0.2577	0.8915	0.1978
Hardness	PLSR	0.9339	0.2573	0.8284	0.2031
Springiness	PLSR	0.8096	0.3924	0.7185	0.4938
Resilience	LS-SVM	0.8865	0.2792	0.5612	0.5218

**Note:** *r*_c_: coefficients of determination for calibration. *r*_p_: coefficients of determination for prediction. RMSEC: root-mean-square error for calibration. RMSEP: root-mean-square error for prediction. PLSR: partial least-square regression. LS-SVM: least-square support vector machines. BP-ANN: backpropagation artificial neural network.

**Table 2 foods-12-03154-t002:** Predicting the pectoral muscle texture profiles using the skin HSI with the reflectance values in the full-wavelength range.

Texture	Best ML Method	*r* _c_	RMSEC	*r* _p_	RMSEP
Gumminess	BP-ANN	0.9812	0.1551	0.9620	0.1792
Chewiness	BP-ANN	0.9810	0.1578	0.9555	0.1709
Cohesiveness	BP-ANN	0.9648	0.0909	0.9306	0.1121
Hardness	PLSR	0.9457	0.2452	0.9033	0.2914
Adhesiveness	PLSR	0.9144	0.2684	0.8895	0.2227
Brittleness	PLSR	0.8958	0.2681	0.8933	0.2044
Resilience	PLSR	0.7934	0.3935	0.6329	0.5168
Springiness	BP-ANN	0.8461	0.3796	0.8140	0.4229

**Table 3 foods-12-03154-t003:** Predicting the abdominal muscle texture profiles using the skin HSI with the reflectance values in the full-wavelength range.

Texture	Best ML Method	*r* _c_	RMSEC	*r* _p_	RMSEP
Chewiness	BP-ANN	0.9786	0.1710	0.9690	0.1356
Gumminess	PLSR	0.9712	0.2005	0.9318	0.1895
Hardness	BP-ANN	0.9682	0.1789	0.9401	0.2337
Adhesiveness	LS-SVM	0.9556	0.1873	0.9206	0.1771
Cohesiveness	PLSR	0.9499	0.1107	0.9213	0.1168
Resilience	PLSR	0.7890	0.4321	0.5798	0.5079
Springiness	PLSR	0.8441	0.3462	0.8294	0.3972
Brittleness	PLSR	0.9061	0.2413	0.8856	0.1891

**Table 4 foods-12-03154-t004:** Predicting the gluteal muscle texture profiles using the skin HSI with the reflectance values in the full-wavelength range.

Texture	Best ML Method	*r* _c_	RMSEC	*r* _p_	RMSEP
Chewiness	BP-ANN	0.9910	0.1015	0.9836	0.1078
Gumminess	PLSR	0.9593	0.2153	0.9234	0.2366
Adhesiveness	BP-ANN	0.9338	0.223	0.9303	0.1907
Cohesiveness	PLSR	0.9269	0.1232	0.8952	0.0931
Hardness	PLSR	0.9069	0.2714	0.8990	0.2926
Brittleness	LS-SVM	0.8782	0.2667	0.8421	0.2171
Resilience	PLSR	0.7835	0.0118	0.6971	0.0124
Springiness	PLSR	0.7513	0.0738	0.6842	0.0865

**Table 5 foods-12-03154-t005:** Predicting the dorsal muscle texture profiles using the reflectance values in the optimal wavelength range.

Texture	ML Method	No. of WLs	*r* _c_	RMSEC	*r* _p_	RMSEP
Gumminess	BP-ANN	88	0.9912	0.1361	0.9847	0.1070
Chewiness	BP-ANN	84	0.9755	0.2450	0.9469	0.2164
Cohesiveness	PLSR	88	0.9714	0.2535	0.9367	0.2836
Hardness	PLSR	86	0.9586	0.2972	0.9298	0.3165
Adhesiveness	BP-ANN	84	0.9432	0.3653	0.9191	0.3004
Brittleness	PLSR	97	0.9112	0.4423	0.8804	0.3848
Springiness	BP-ANN	110	0.8380	0.5171	0.7194	0.7554
Resilience	BP-ANN	80	0.8346	0.5676	0.6493	0.7074

**Note:** No. of WLs: number of wavelengths.

**Table 6 foods-12-03154-t006:** Predicting the pectoral muscle texture profiles using the reflectance values in the optimal wavelength range.

Texture	ML Method	No. of WLs	*r* _c_	RMSEC	*r* _p_	RMSEP
Gumminess	BP-ANN	67	0.9883	0.1605	0.9574	0.2439
Chewiness	BP-ANN	60	0.9862	0.1771	0.9552	0.2285
Adhesiveness	LS-SVM	75	0.9424	0.3643	0.9370	0.2499
Cohesiveness	PLSR	88	0.9556	0.3081	0.9056	0.3765
Hardness	BP-ANN	69	0.9379	0.3561	0.9033	0.3930
Brittleness	PLSR	75	0.8835	0.5213	0.8846	0.3753
Springiness	LS-SVM	77	0.8866	0.4490	0.7376	0.6971
Resilience	LS-SVM	81	0.7095	0.6983	0.5964	0.7960

**Table 7 foods-12-03154-t007:** Predicting the abdominal muscle texture profiles using the reflectance values in the optimal wavelength range.

Texture	ML Method	No. of WLs	*r* _c_	RMSEC	*r* _p_	RMSEP
Chewiness	BP-ANN	77	0.9858	0.1779	0.9776	0.1649
Gumminess	PLSR	81	0.9740	0.2418	0.9517	0.2360
Hardness	PLSR	92	0.9592	0.2860	0.9392	0.3158
Cohesiveness	PLSR	75	0.9631	0.2859	0.9070	0.3601
Adhesiveness	LS-SVM	88	0.9564	0.3245	0.8623	0.3358
Brittleness	BP-ANN	98	0.9309	0.4130	0.8617	0.3860
Springiness	PLSR	96	0.8954	0.4320	0.8322	0.5928
Resilience	LS-SVM	84	0.9367	0.3525	0.5609	0.7414

**Table 8 foods-12-03154-t008:** Predicting the gluteal muscle texture profiles using the reflectance values in the optimal wavelength range.

Texture	ML Method	No. of WLs	*r* _c_	RMSEC	*r* _p_	RMSEP
Chewiness	BP-ANN	74	0.9768	0.2174	0.9768	0.1804
Gumminess	BP-ANN	70	0.9614	0.2804	0.9339	0.2856
Adhesiveness	BP-ANN	85	0.9421	0.3614	0.9304	0.2861
Cohesiveness	LS-SVM	79	0.9826	0.1891	0.8726	0.3938
Hardness	PLSR	80	0.9092	0.4186	0.8486	0.4933
Brittleness	PLSR	74	0.8883	0.4723	0.7613	0.4921
Springiness	PLSR	114	0.7668	0.6437	0.6396	0.7508
Resilience	PLSR	89	0.7066	0.7431	0.6316	0.7230

## Data Availability

The data presented in this study are available on request from the corresponding author. The data presented in this study are available in Appendix A.

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
