# Peer review of "Rapid and Non-Invasive Assessment of Texture Profile Analysis of Common Carp (Cyprinus carpio L.) Using Hyperspectral Imaging and Machine Learning"

_foods, 2023, doi:10.3390/foods12173154_

Round 1
Reviewer 1 Report
It is a good and practical study but I made the following observations which should be addressed by the authors. After doing the following points, it can be accepted in the Journal of Foods.
- It is recommended that it be edited by an English expert.
Abstract
- Its ok.
Keywords
- Sort by alphabetical order.
- It is better to use keywords other than the title.
Introduction
Introduction
The logic of the current introduction should be revised, and I suggest organizing the Introduction section as following order: importance and meanings, previous studies (literature review), the gaps of previous studies, and objectives of this study.
The novelty of the research is not justified.
Introduction is not including up to date and relevant articles
Please see the following articles and to make the article more complete, they must be added to the text
Oxidative stability of virgin olive oil: evaluation and prediction with an adaptive neuro-fuzzy inference system (ANFIS)
Materials and methods
- Add references to all sections that are needed.
Result
- It is necessary to give a brief explanation in the text for each figure and table (especially tables and supplementary figures).
-It is necessary to define all abbreviations in tables and figures so that they can be understood without referring to the text.
Discussion
- The discussion section is very weak. It is necessary to compare with more studies and also to state the possible reasons for all the obtained results (in all tables and figures) by mentioning the reference.
- If possible, express the results of previous studies numerically.
- Add references wherever you make a claim or argument.
Conclusion
-Finally, state your limitations of the study and possible suggestions for other researchers.
It is recommended that it be edited by an English expert.
Author Response
Detailed Responses to Referees’ Comments from Referees
Reviewer #1:
It is a good and practical study but I made the following observations which should be addressed by the authors. After doing the following points, it can be accepted in the Journal of Foods.
Question 1: Sort by alphabetical order.
Reply: Thanks very much for your valuable suggestions. We have sorted the keywords alphabetically (line 30).
Question 2: It is better to use keywords other than the title.
Reply: We are grateful for the suggestion. We have corrected the keywords as your suggestions (line 30).
Question 3: The logic of the current introduction should be revised, and I suggest organizing the Introduction section as following order: importance and meanings, previous studies (literature review), the gaps of previous studies, and objectives of this study.
Reply: Thank you very much for the reviewers' valuable comments on the logic of the introduction in our manuscript. We strongly agree with you that perhaps our presentation was not very clear, for which we apologize. We have revised the introduction section, especially with regard to the gaps of previous studies, and objectives of this study (section 1).
Question 4: The novelty of the research is not justified.
Reply: We deeply apologized for the unclear statement regarding the novelty of this study. The novelties of this study mainly include (1) we used hyperspectral imaging at 400-1000 nm to acquire hyperspectral image data of the epidermis of living common carp rather than muscle mass to assess the textural parameters of the muscle [1-3], (2) six rather than one or two models were used to compare the effects of predicting texture parameters in different muscle regions, and (3) the distribution of texture parameters of the common carp muscle was visualized. We have deliberately emphasized the novelty of this study in the introduction section.
Question 5: Introduction is not including up to date and relevant articles
Reply: Following the reviewer's suggestion, we have added references from the last three years in relevant fields in the introduction section (line 43 and line 52).
Question 6: Please see the following articles and to make the article more complete, they must be added to the text. Oxidative stability of virgin olive oil: evaluation and prediction with an adaptive neuro-fuzzy inference system (ANFIS)
Reply: According to the reviewer's suggestion, we have revised the article to be completer and more added this reference to the introduction (line 56).
Question 7: Add references to all sections that are needed.
Reply: Following the reviewer’s suggestion, we have added references for all sections that are needed (section 1).
Question 8: It is necessary to give a brief explanation in the text for each figure and table (especially tables and supplementary figures).
Reply: Thanks very much for your valuable suggestions. We have briefly described and annotated each figure and table in the manuscript.
Question 9: It is necessary to define all abbreviations in tables and figures so that they can be understood without referring to the text.
Reply: According to the reviewer’s suggestion, we have defined all abbreviations appearing in the tables so that they could be understood without referring to the text (line 482, line 487 to line 489, line 492 to line 495, line 498 to line 501, line 548 to line 551, line 554 to line 557, line 577 to line 580, line 583 to line 586).
Question 10: The discussion section is very weak. It is necessary to compare with more studies and also to state the possible reasons for all the obtained results (in all tables and figures) by mentioning the reference.
Reply: We have made correction according to the reviewer’s comments. We have discussed the spectral differences between the different muscle regions of common carp that emerged from the results. We have provided an in-depth discussion of the results obtained, including the potential mechanisms behind the results (Section 4).
Question 11: If possible, express the results of previous studies numerically.
Reply: As Reviewer suggested that we have listed the predictive coefficients of the model numerically when citing the results of previous studies in the discussion section (line 651, line 652, line 661 to line 663, and line 677 to line 678).
Question 12: Add references wherever you make a claim or argument.
Reply: Thank you for the suggestion. We have added references in the discussion section of the manuscript to support our arguments.
Question 13: Finally, state your limitations of the study and possible suggestions for other researchers.
Reply: Following the reviewer’s suggestion, we have stated the limitations of this study and suggested future research directions for other researchers (line 767 to line 772).
Question 14: It is recommended that it be edited by an English expert.
Reply: Thanks for suggesting improvements to the language, the writing skill has been improved. Our manuscripts have been thoroughly edited in English by MDPI (English editing ID: English-69828). We feel sorry for causing you unnecessary troubles in reviewing, we hope that the revised version may meet your exceptions.
Reference
- Cheng, K., Wagner, L., Moazzami, A., Gómez-Requeni, P., Schiller Vestergren, A., Brännäs, E., Pickova, J., & Trattner, S. Decontaminated fishmeal and fish oil from the Baltic Sea are promising feed sources for Arctic char (Salvelinus alpinus)—studies of flesh lipid quality and metabolic profile. Eur. J. Lipid Sci. Technol. 2016, 118, 862-873. DOI: 10.1002/ejlt.201500247.
- Zhou, J., Wu, X., Chen, Z., You, J., & Xiong, S. Evaluation of freshness in freshwater fish based on near infrared reflectance spectroscopy and chemometrics. LWT 2019, 106, 145-150; DOI: 10.1016/j.lwt.2019.01.056.
- Moosavi-Nasab, M., Khoshnoudi-Nia, S., Azimifar, Z., & Kamyab, S. Evaluation of the total volatile basic nitrogen (TVB-N) content in fish fillets using hyperspectral imaging coupled with deep learning neural network and meta-analysis. Rep. 2021, 11, 5094. DOI: 10.1038/s41598-021-84659-y.

Reviewer 2 Report
In this study, the authors evaluated the texture profile of four muscle regions living common carp muscles by scanning the corresponding skin regions with Hyperspectral imaging (HSI). They collected the skin hyperspectral information of four regions of 387 scaled and live common carp. Eight texture indicators of the muscle corresponding to each skin region were measured. With the skin HSIs of live common carp, six machine learning (ML) models were used to predict the muscle texture indicators. Back propagation artificial neural network (BP-ANN), partial least squares regression (PLSR), and least-squares support vector machines (LS-SVM) were identified as the optimal models to predict the texture parameters of the dorsal (coefficients of determination for prediction (rP) ranged from 0.9191 to 0.9847, and the root mean square error for prediction ranged from 0.1070 to 0.3165), pectoral (rP) in the range of 0.9033 to 0.9574, and RMSEP ranged from 0.2285 to 0.3930), abdominal (rP) ranged from 0.9070 to 0.9776, and RMSEP ranged from 0.1649 to 0.3601), and gluteal (rP) ranged from 0.8726 to 0.9768, and RMSEP ranged from 0.1804 to 0.3938) regions. The optimal ML models and skin HSI data were employed to generate the visual prediction maps of TPA values in common carp muscles. These results demonstrated that skin HSI and the optimal models would be used to rapidly and accurately determine the texture qualities of different muscle regions in common carp.
1. What are the key features of the proposed schemes? (Properties, characteristics, and weaknesses).
2. Keywords must not include any word existed in the title.
3. Punctuation marks should be checked throughout the manuscript, especially after the equations.
4. The manuscript has many abbreviations and acronyms. Please list them with descriptions in a table in the manuscript.
5. A step-by-step algorithm to describe the technique will be beneficial.
6. Kindly list the CPU time or memory comparisons with other results in Section 3.
7. Please mention which challenges you face to obtain the results.
8. Please add a section for future recommendations.
Author Response
Detailed Responses to Referees’ Comments from Referees
Reviewer #2:
In this study, the authors evaluated the texture profile of four muscle regions living common carp muscles by scanning the corresponding skin regions with Hyperspectral imaging (HSI). They collected the skin hyperspectral information of four regions of 387 scaled and live common carp. Eight texture indicators of the muscle corresponding to each skin region were measured. With the skin HSIs of live common carp, six machine learning (ML) models were used to predict the muscle texture indicators. Back propagation artificial neural network (BP-ANN), partial least squares regression (PLSR), and least-squares support vector machines (LS-SVM) were identified as the optimal models to predict the texture parameters of the dorsal (coefficients of determination for prediction (rP) ranged from 0.9191 to 0.9847, and the root mean square error for prediction ranged from 0.1070 to 0.3165), pectoral (rP) in the range of 0.9033 to 0.9574, and RMSEP ranged from 0.2285 to 0.3930), abdominal (rP) ranged from 0.9070 to 0.9776, and RMSEP ranged from 0.1649 to 0.3601), and gluteal (rP) ranged from 0.8726 to 0.9768, and RMSEP ranged from 0.1804 to 0.3938) regions. The optimal ML models and skin HSI data were employed to generate the visual prediction maps of TPA values in common carp muscles. These results demonstrated that skin HSI and the optimal models would be used to rapidly and accurately determine the texture qualities of different muscle regions in common carp.
Question 1: What are the key features of the proposed schemes? (Properties, characteristics, and weaknesses).
Reply: As mentioned in our manuscript, our study aims to develop a non-destructive method combining skin hyperspectral imaging and machine learning to detect textual features in live fish muscles. In contrast to other previous studies, we accelerated the textural features of fish muscle by scanning epidermal hyperspectral images of live carp instead of muscle mass. Although the prediction coefficients were not found to be very high for some texture parameters in our study, desirable results were achieved for most of the desirable, demonstrating the potential of the method.
Question 2: Keywords must not include any word existed in the title.
Reply: We are grateful for the suggestion. We have corrected the keywords as your suggestions (line 30).
Question 3: Punctuation marks should be checked throughout the manuscript, especially after the equations.
Reply: Thanks for this valuable comment. According to the reviewer's suggestions we checked the punctuation throughout the manuscript and corrected any incorrect punctuation.
Question 4: The manuscript has many abbreviations and acronyms. Please list them with descriptions in a table in the manuscript.
Reply: Thanks very much for your valuable suggestions. In order to keep the manuscript concise and clear, we have defined the abbreviations that appear for the first time in the manuscript. As reviewer' s suggestion, we checked the occurrence of abbreviations and acronyms and labeled them in the manuscript (line 482, line 487 to line 489, line 492 to line 495, line 498 to line 501, line 548 to line 551, line 554 to line 557, line 577 to line 580, line 583 to line 586).
Question 5: A step-by-step algorithm to describe the technique will be beneficial.
Reply: We agree with the reviewer that A step-by-step algorithm to describe the technique will be beneficial to the present study. However, the aim of our study was to screen six predictive models for their effect on the prediction of common carp muscle texture parameters. Considering the length of the article and the fact that these models have been mentioned in previous studies. Therefore, we provided a brief overview of the algorithms of the models in the section 2.6 and inserted the references.
Question 6: Kindly list the CPU time or memory comparisons with other results in Section 3.
Reply: Thank you for your attention to the CPU time or memory in our study, you must be an expert in this field.However, we believe that CPU time or memory would be outside the scope of our paper, because we found that the accuracy of the model as well as its stability is the core issue of the study by comparing it with other similar studies. We also reviewed the literature and did not see any mention of CPU time and memory related information in it [1-3].
Question 7: Please mention which challenges you face to obtain the results.
Reply: Considering the Reviewer’s suggestion, we have listed the challenges to achieving the results (line 674 to 683).
Question 8: Please add a section for future recommendations.
Reply: As Reviewer suggested that we emphasized the potential of our research in the field in the future (line 769 to line 771).
Reference
- León-Ecay, S.; López-Maestresalas, A.; Murillo-Arbizu, M.; Beriain, M.; Mendizabal, J.; Arazuri, S.; Jarén, C.; Bass, P.; Colle, M.; García, D.; Romano-Moreno, M.; Insausti, K. Classification of Beef longissimus thoracis Muscle Tenderness Using Hyperspectral Imaging and Chemometrics. Foods 2022, 11, 3105. DOI: 10.3390/foods11193105.
- Liu, C.; Chu, Z.; Weng, S.; Zhou, G.; Han, K.; Zhang, Z.; Huang, L.; Zhou, Z.; Zheng, S. Fusion of electronic nose and hyperspectral imaging for mutton freshness detection using input-modified convolution neural network. Food Chemistry 2022, 385, 132651-. DOI: 10.1016/j.foodchem.2022.132651.
- Pu, H.; Yu, J.; Sun, D.; Wei, Q.; Shen, X.; Wang, Z. Distinguishing fresh and frozen-thawed beef using hyperspectral imaging technology combined with convolutional neural networks. J. 2023, 189, 108559. DOI: 10.1016/j.microc.2023.108559.

Reviewer 3 Report
In my opinion, the manuscript should contain all the information that serves the idea and makes it clear to the reader Therefore; I find that there is a lot of important information that should be included in the manuscript.
1. Where is the composition of the commercial feed used in feeding carp and its components?
2. What about the average weight of these fish?
3. Was this method based on prediction compared to the traditional method?
4. Is this method limited to this type of carp fish?
5. What is the number of experimental replicates in each experimental treatment?
6. English should improve by a native person. The paper suffers from a poor English structure throughout and cannot be published or reviewed properly in the current format. The manuscript requires a thorough proofread by a native person whose first language is English.
7. The novelty of the study needs to be highlighted compared to other similar studies.
8. Discussion is weak. The discussion needs enhancement with real explanations not only agreements and disagreements. Authors should improve it by the demonstration of biochemical/physiological causes of obtained results. Instead of justifying results, results should be interpreted, explained to appropriately elaborate inferences. discussion seems to be poor, didn't give good explanations of the results obtained. I think that it must be really improved. Where possible please discuss potential mechanisms behind your observations.
9. You should also expand the links with prior publications in the area but try to be careful not to over-reach. For the latter, you should highlight potential areas of future study.
10. A detailed "Conclusion" should be provided to state the final result that the authors have reached.
11. Please note you only need to place your conclusion and not keep putting results, because these have already been presented in the manuscript.
12. Author(s) should re-format the references based on journal format. See the instructions for authors.
English should improve by a native person. The paper suffers from a poor English structure throughout and cannot be published or reviewed properly in the current format. The manuscript requires a thorough proofread by a native person whose first language is English.
Author Response
Detailed Responses to Referees’ Comments from Referees
Reviewer #3:
In my opinion, the manuscript should contain all the information that serves the idea and makes it clear to the reader Therefore; I find that there is a lot of important information that should be included in the manuscript.
Question 1: Where is the composition of the commercial feed used in feeding carp and its components?
Reply: In the revised section 2.1. Ethics Statement and Sampling, we have added the ingredients of the commercial diet used in feeding common carp to the manuscript (line 97 to line 100).
Question 2: What about the average weight of these fish?
Reply: The average weight of these fish is 607.09g. We have added it in Section 2.1 (line 94).
Question 3: Was this method based on prediction compared to the traditional method?
Reply: We apologize for not explaining clearly in the manuscript this issue raised by the reviewer. The method used in this study for the detection of texture values in different muscle regions of living common carp is based on prediction. The optimal prediction model is determined by dividing the samples into a correction set and a prediction set, which enables the prediction of texture results by scanning the spectral information from the surface of the live common carp.
Question 4: Is this method limited to this type of carp fish?
Reply: We thank the reviewer for the very interesting comment. This method has only been applied to common carp in the present study. In fact, common carp was chosen as the subject in our study. The reason for choosing common carp is that common carp is an important freshwater aquaculture species in China. In the future, it is possible to apply this method to other species of fish in the future. Therefore, we have highlighted the potential areas of future study (line 770 to line 772).
Question 5: What is the number of experimental replicates in each experimental treatment?
Reply: A total of 387 common carp were collected as experimental subjects in our study, and the hyperspectral data as well as the texture values of different muscle regions were acquired separately. The 387 datasets were divided into a calibration set and a validation set for constructing predictive models of muscle texture indicators in live common carp.
Question 6: English should improve by a native person. The paper suffers from a poor English structure throughout and cannot be published or reviewed properly in the current format. The manuscript requires a thorough proofread by a native person whose first language is English.
Reply: Thanks for suggesting improvements to the language, the writing skill has been improved. Our manuscripts have been thoroughly edited in English by MDPI (English editing ID: English-69828). We feel sorry for causing you unnecessary troubles in reviewing, we hope that the revised version may meet your exceptions.
Question 7: The novelty of the study needs to be highlighted compared to other similar studies.
Reply: Thank you for your feedback and raising the concern regarding the novelty of our study. We appreciate your attention to previously published studies in the field. As we mentioned in the manuscript, we acquired hyperspectral image information of the skin of living common carp rather than that of the muscle mass. Compared to other similar previous studies, our findings are more rapid, efficient and non-invasive to assess of fish muscle texture profile in the future. We have emphasized the novelty and research objectives of this study in the introduction section (line 83 to line 89). We hope this clarification addresses your concerns regarding the novelty of our study.
Question 8: Discussion is weak. The discussion needs enhancement with real explanations not only agreements and disagreements. Authors should improve it by the demonstration of biochemical/physiological causes of obtained results. Instead of justifying results, results should be interpreted, explained to appropriately elaborate inferences. discussion seems to be poor, didn't give good explanations of the results obtained. I think that it must be really improved. Where possible please discuss potential mechanisms behind your observations.
Reply: Thank you for underlining this deficiency. Following the reviewers' comments, we have made significant changes to the discussion section. We have discussed the spectral differences between the different muscle regions of common carp that emerged from the results. We have provided an in-depth discussion of the results obtained, including the potential mechanisms behind the results (Section 4).
Question 9: You should also expand the links with prior publications in the area but try to be careful not to over-reach. For the latter, you should highlight potential areas of future study.
Reply: We have made correction according to the Reviewer’s comments. We have strengthened links to previous research in the field by including references in the discussion section (line 649 to line 655, line 658 to line 664, line 673 to line 682). And we emphasized the potential of our research in the field in the future (Section 5).
Question 10: A detailed "Conclusion" should be provided to state the final result that the authors have reached.
Reply: Thank you for the above suggestion. Following your suggestion, we have modified the conclusion section to obtain conclusion to make the entire findings of this study more visual and detailed.
Question 11: Please note you only need to place your conclusion and not keep putting results, because these have already been presented in the manuscript.
Reply: Thanks for this valuable comment. We have revised the conclusions according to the reviewer's suggestions, drawing conclusions based on the results available, presented the limitations of this study and emphasized the areas for future research (line 765 to line 770).
Question 12: Author(s) should re-format the references based on journal format. See the instructions for authors.
Reply: Considering the Reviewer’s suggestion, we have re-formatted the references based on journal format (line 832 to line 1276).
Question 13: English should improve by a native person. The paper suffers from a poor English structure throughout and cannot be published or reviewed properly in the current format. The manuscript requires a thorough proofread by a native person whose first language is English.
Reply: Our manuscripts have been thoroughly edited in English by MDPI (English editing ID: English-69828). We feel sorry for causing you unnecessary troubles in reviewing, we hope that the revised version may meet your exceptions.

Round 2
Reviewer 1 Report
Accept